# Adipokines and Autoimmunity in Inflammatory Arthritis

**DOI:** 10.3390/cells10020216

**Published:** 2021-01-22

**Authors:** Elena Neumann, Rebecca Hasseli, Selina Ohl, Uwe Lange, Klaus W. Frommer, Ulf Müller-Ladner

**Affiliations:** Department of Rheumatology and Clinical Immunology, Campus Kerckhoff, Justus Liebig University Giessen, 61231 Bad Nauheim, Germany; r.hasseli@kerckhoff-klinik.de (R.H.); s.ohl@kerckhoff-klinik.de (S.O.); U.lange@kerckhoff-klinik.de (U.L.); K.frommer@kerckhoff-klinik.de (K.W.F.); u.mueller-ladner@kerckhoff-klinik.de (U.M.-L.)

**Keywords:** adipokines, adipocytokines, rheumatic diseases, rheumatoid arthritis, autoimmunity

## Abstract

Adipokines are adipose tissue-derived factors not only playing an important role in metabolism but also influencing other central processes of the body, such as inflammation. In autoimmune diseases, adipokines are involved in inflammatory pathways affecting different cell types. Many rheumatic diseases belong to the group of autoimmune diseases, for example rheumatoid arthritis (RA) and psoriatic arthritis. Due to the autoimmune responses, a chronic inflammatory milieu develops, which affects the whole body, including adipose tissue. Metabolic alterations such as obesity influence inflammatory responses in autoimmune diseases. Adipokines are bioactive mediators mainly produced by adipose tissue. Due to alterations of systemic adipokine levels, their role as biomarkers with diagnostic potential has been suggested in the context of rheumatic diseases. In the affected joints of RA patients, different synoviocytes but also osteoclasts, osteoblasts, and chondrocytes produce several adipokines, contributing to the unique inflammatory microenvironment. Adipokines have been shown to be potent modulatory effectors on different cell types of the immune system but also local cells in synovial tissue, cartilage, and bone. This review highlights the most recent findings on the role of adipokines in the pathophysiology of inflammatory arthritis with a distinct focus on RA in the quickly developing research field.

## 1. Introduction

Disorders affecting the joint can be divided in two central groups. Primary inflammatory autoimmune arthritides consist of rheumatoid arthritis (RA) and spondyloarthritides, including psoriatic arthritis (PsA) and ankylosing spondylitis. Findings in inflammatory arthritis are often compared to non-autoimmune osteoarthritis (OA). OA is usually induced by previous joint injury, biomechanical stress on joints due to malposition, or overweight and obesity. Metabolic diseases, such as diabetes or hyperlipidemia, are considered key risk factors for OA development [1]. Though cartilage degradation is a central feature of OA, resulting in an inflammatory response due to mechanical joint damage, the whole joint is affected, including adjacent bone, which involves the formation of osteophytes [1]. In contrast, RA is an autoimmune disorder characterized by severe chronic inflammation early within the disease, leading to irreversible joint damage if left untreated [2]. RA affects mainly peripheral joints symmetrically, leading to progressive joint damage. The clinical presentation of RA varies, and environmental factors and epigenetic mechanisms have been associated with this disorder [3,4]. A persistent inflammatory infiltration of the synovial tissue contributes to local synovial cell activation, which causes the release of pro-inflammatory and matrix-degrading factors to the tissue and joint space. RA synovial fibroblasts (RASFs) exhibit an autonomous activated phenotype contributing to the inflammatory milieu as well as synovial hyperplasia [4,5].

PsA and RA are both rheumatic chronic inflammatory diseases sharing similarities such as synovitis but also differences in the pathophysiology. This includes a different vascular pattern in the affected joints [6], a less pronounced synovial hyperplasia in PsA compared to RA patients [7], and the cell infiltrates at the inflamed synovial–entheseal complex being more prominent in PsA patients [8]. Therefore, enthesitis is one of the frequent features of PsA according to the Classification of Psoriatic Arthritis (CASPAR) criteria [9,10] but rare in the case of RA as well as OA. PsA patients also respond differently to treatment. For example, the anti-IL17A biologic secukinumab is more effective in PsA than in RA patients.

Dysregulation of immune-endocrine circuits is involved in the development of chronic metabolic disorders, such as obesity, metabolic syndrome, and diabetes, but also plays a role in chronic inflammatory diseases, such as RA [11]. Understanding the mechanisms of both immune regulation and resolution of inflammation is crucial for the development of successful treatment approaches to achieve and maintain remission or low disease activity in RA and PsA. Regulatory mechanisms include mediators such as cytokines, chemokines, hormones, neurotransmitters, and their receptors from inflamed tissue but also the periphery, including the adipose tissue. Many diseases are based on chronic inflammation lasting for years or for the patients’ whole life if left untreated. Due to alterations of tissues or tissue components in the course of the disease as well as specifically due to autoimmune responses, the organism is no longer able to terminate the pathophysiological inflammatory circle, resulting in continuous tissue damage and, finally, loss of function and quality-of-life impairment.

## 2. Adipose Tissue and Adipokines

Adipose tissue is the key tissue regulating energetic homeostasis. It also serves as an endocrine organ due to secretion of a large number of bioactive substances. The factors that are released by adipose tissue are called adipocytokines. This group of mediators includes adipokines, cytokines, chemokines, complement factors, and hormones [12]. Cytokines produced by an excess of adipose tissue can affect the whole body, leading to the development of a so-called low-level inflammation, which can be observed in obese individuals.

Adipokines are factors mainly produced by adipocytes in the white adipose tissue. Adipose tissue secretes a large number of highly biologically active factors. However, many adipokines such as leptin and adiponectin are also known modulators of immune responses. Systemic alterations of adipokines have been identified for a large number of chronic inflammatory diseases, and the potential of adipokines such as adiponectin and leptin has been discussed. Therefore, adipokines have been investigated for many years in the context of chronic inflammatory and degenerative rheumatic diseases systemically as well as on the local level in cells and tissues. Furthermore, pro- and anti-inflammatory properties of adipokines have been identified. Therefore, it has been accepted in the past years that adipokines play an important role in immune-mediated rheumatic disease and degenerative OA.

### 2.1. Adiponectin

Adiponectin, encoded by the *ADIPOQ* gene, has been described as a mainly anti-inflammatory adipokine. Adiponectin is produced in large amounts by adipocytes of the white adipose tissue [13]. Adiponectin concentrations are inversely correlated with the body mass index (BMI). Adiponectin is a complex molecule. Adiponectin monomers form different isoforms, depending on the degree of oligomerization: the trimer (low molecular weight (LMW)), the hexamer (middle molecular weight (MMW)), and the multimeric (high molecular weight (HMW)) adiponectin consisting of 12–18 monomers. Globular adiponectin, consisting of the monomeric head-domain, is formed by proteolytic cleavage. The monomer occurs as an intermediate in adipocytes, while in the circulation the main forms are the multimeric adiponectin isoforms. Besides the two main adiponectin receptors, AdipoR1 and AdipoR2, which are able to bind globular and full-length adiponectin isoforms with different affinities, other receptors such as T-cadherin and PAQR3 (progestin and AdipoQ receptor family member 3) have been described [12,13,14,15,16,17]. Recently, Tanaka and colleagues showed that adiponectin overexpression increased the regeneration of myofibers promoting muscle regeneration in a T-cadherin-dependent manner [17]. Adiponectin has several central biological functions, such as fatty acid biosynthesis and inhibition of gluconeogenesis within the liver [12,13,15]. However, adiponectin not only shows potential as a biomarker due to the systemic alterations under inflammatory condition, which were found to decrease [18,19,20] or increase [21,22,23] in physiological and pathophysiological conditions, but is also actively involved in inflammatory responses and affects different cell types. In type 2 diabetes, atherosclerosis, and metabolic syndrome, predominantly anti-inflammatory effects have been described [12,18,24]. However, in the context of rheumatoid arthritis, the role of adiponectin is not fully understood. On cellular and tissue level, opposite effects have been described for rheumatic diseases, such as RA, where it seems to promote inflammation and tissue damage in the affected joints. Recent findings will be outlined in the sections below.

### 2.2. Leptin

The main adipokine produced by adipocytes is leptin. Leptin concentrations are positively correlated with white adipose tissue mass. In addition, leptin has central functions in metabolism and also plays a role in inflammation and inflammatory disorders [25]. Leptin is encoded by the *LEP (ob)* gene and is a 16 kDa non-glycosylated protein. The effects of leptin are mediated by binding to the long form of the leptin receptor LEPR [26]. By inducing anorexigenic factors, leptin is known to be a central protein in appetite regulation and obesity. However, leptin is also involved in many processes besides insulin secretion and basal metabolism, such as reproduction, bone mass regulation, and (chronic) inflammatory diseases [14,25]. Leptin, in turn, is induced in adipose tissue, depending on the energy status, by sex hormones and by inflammatory mediators [14,25]. In contrast to adiponectin, leptin is considered a pro-inflammatory adipokine. It is involved in low-grade inflammation due to overweight in obesity [26]. Both the innate and adaptive immune responses are affected by leptin, and the LEPR is expressed at the surface of most immune cells. Leptin increases the phagocytosis of macrophages, induces proliferation of monocytes and macrophages, alters the cytotoxicity of natural killer cells as well as the proliferation of CD4 T cells, suppresses type 2 T helper cells (Th2), and increases Th1 responses as well as regulatory T cell (Treg) responses [25,27]. Leptin itself is induced by pro-inflammatory cytokines during acute infection and sepsis, but it is also induced during chronic inflammatory autoimmune diseases [25,28].

### 2.3. Visfatin

Visfatin, or pre-B-cell colony-enhancing factor (PBEF), is a multifunctional protein that has the ability to promote B cell differentiation and possesses nicotinamide phosphoribosyl-transferase (Nampt) enzymatic activity. These different mechanisms of action make visfatin a very interesting protein. Whether altered local visfatin levels are associated with its extracellular interaction with cells or due to its intracellular enzymatic Nampt activity, leading to changes in the nicotinamide adenine dinucleotide content, remains to be clarified. Visfatin is produced by adipose tissue as well as other tissues, such as liver, bone marrow, and muscle. It can be induced by pro-inflammatory factors, chemokines, hypoxia, and visfatin itself, and in turn visfatin induces a pro-inflammatory response in many cell types and tissues by itself [12,15,24]. The cell surface receptor for visfatin is unknown, and several studies have shown that the visfatin effects are in part due to its Nampt activity [14]. However, an interaction with insulin-like growth factor (IGF)-1 signaling has been reported. Here, it can been shown that visfatin inhibits IGF-1-mediated function independently of the IGF-1 receptor activation [29], suggesting another mechanism of action besides its Nampt activity. Interestingly, it has been recently observed that several micro RNAs (miRNAs) are involved in the visfatin-mediated effects specifically in the context of OA chondrocytes [30]. Visfatin significantly reduced viability and induced apoptosis in OA chondrocytes, involving the NFκB pathway as well as decreasing miR-140 and miR-146a and increasing iR-let7e expression in this study [30].

### 2.4. Resistin

Resistin is a homodimeric cysteine-rich protein that is mainly produced by macrophages in humans, whereas in animals such as mice, the source of resistin is the adipose tissue resident adipocytes [14,31]. Therefore, results from animal studies can often not directly be translated to the human situation. Human resistin was described to be a mainly pro-inflammatory protein promoting immune cell recruitment and immune cell activation [14,32] because it is produced by macrophages. Furthermore, a role in the context of the development of, for example, coronary artery disease, atherosclerosis, type 2 diabetes, as well as OA and psoriasis has been reported [14,33], showing a mainly inflammation-promoting effect. However, anti-inflammatory effects of resistin have also been described on cellular and tissue level [31]. Therefore, the effect of resistin may depend on the tissue and pathophysiological condition studied. In addition, an immunomodulatory role in the context of rheumatic diseases, including RA, PsA, and OA, has been described [14].

### 2.5. Chemerin, Vaspin, and Omentin

Chemerin precursors consist of a hydrophobic signal peptide sequence, a cysteine fold-containing domain, and a labile C domain. Chemerin is activated after hydrolization by cysteine or serine proteases. Removal of the signal sequence leads to a secreted preform with low biological activity. Different cleaved isoforms and underlying mechanisms of cleavage have been described in the past years [34], which are in the focus of current investigations. Chemerin is involved in immune responses, and its role in coronary atherosclerosis and metabolic syndrome development, among other diseases, has been described [14]. Mainly anti-inflammatory chemerin properties have been described for macrophages as well as in an LPS-induced acute lung injury mouse model [35]. On the cellular level, chemerin acts as a chemoattractant for natural killer cells, macrophages, and dendritic cells [14,36].

Vaspin belongs to a family of serine protease inhibitors. Vaspin has been associated with insulin resistance and metabolic syndrome as well as atherosclerosis and cardiovascular disease [37]. Besides subcutaneous adipose tissue, vaspin is also expressed and produced in other tissues such as skin and skeletal muscle. Interestingly, vaspin has been described to be involved in skeletal muscle inflammation [38]. Vaspin overexpression in mice showed altered metabolism and inflammation with improved glucose tolerance and resistance to high-fat diet-induced obesity with lower systemic IL-6 levels [14]. On the cellular level, vaspin alters adipocyte differentiation and glucose homeostasis. In the context of coronary atheromatous plaques, the pro-inflammatory phenotype of macrophages was suppressed by vaspin [14]. However, limited knowledge regarding the evaluation of the specific role of vaspin in the context of autoinflammation in RA is available.

Omentin is a glycoprotein that binds to galactofuranosyl residues of microorganisms and to the lactoferrin-binding protein. It is expressed mainly in omental adipose tissue. However, omentin is abundant in the plasma of healthy donors [39]. Omentin has been described to mainly have anti-inflammatory effects. Anti-atherogenic effects in obese individuals have been described, as well as a negative association with inflammatory bowel disease and metabolic syndrome [14,39].

### 2.6. Progranulin, Lipocalin-2, and Nesfatin

Progranulin (PGRN, granulin/epithelin precursor) consists of seven granulin/epithelin repeats that can be cleaved into small homologous subunits. Full-length protein as well as the resulting peptides after cleavage are biologically active [40]. PGRN is an autocrine factor promoting different physiological processes, such as chondrocyte differentiation and proliferation as well as enchondral ossification [14]. Pro- and anti-inflammatory effects of PGRN have been described. The anti-inflammatory properties are mainly mediated by competitive binding to tumor necrosis factor (TNF) receptors disturbing the TNFα-mediated responses [25,41]. PGRN is produced by many different cells such as adipocytes, macrophages, and chondrocytes and has been suggested as a potential biomarker in inflammatory disease [42].

Nesfatin (nesfatin-1) is an adipokine involved in satiety induction and in energy homeostasis. It is secreted by the hypothalamus and acts as an anorexigenic factor. In addition, it is produced by subcutaneous adipose tissue and other tissues within the gut, pancreas, and testes [43]. Lipocalin-2 (LCN2) is an adipokine induced by pro-inflammatory factors, such as IL-1beta, LPS, and other cytokines, as well as dexamethasone and other adipokines, such as leptin and adiponectin [14,44,45].

## 3. Adipokines in Autoimmune Rheumatoid Arthritis

### 3.1. Rheumatoid Arthritis

RA affects 1% of the population worldwide, leading to loss of physical joint function. It is a systemic chronic autoimmune disease developing in genetically susceptible individuals due to environmental factors and involving epigenetic mechanisms. Being a very heterogeneous disorder with different clinical forms and dominant pathomechanisms, treatment of RA still remains challenging [2,3,4,13]. Regarding rheumatic diseases, RA was the first to be evaluated in the context of adipokines, and their systemic levels are mostly compared with those in primarily non-inflammatory degenerative osteoarthritis and healthy donors. Due to the limited availability of control cells and tissues from healthy donors, findings on the cellular level are usually compared to cells from OA patients.

Many different cells are involved in the pathogenesis of chronic inflammatory joint diseases such as RA. Friction-free joint mobility is ensured by the synovial membrane and synovial fluid. The synovial membrane consists of a thin layer at the border to the synovial fluid with two main cell types, the macrophage-like synovial cells (called type A) and fibroblast-like synoviocytes (called type B). Both cell types are also present in the sublining, which consists mainly of connective tissue and adipose tissue in the deeper layers of the sublining. Synovial macrophages consist of resident cells but also originate from blood monocytes. Synovial fibroblasts produce hyaluronic acids and proteoglycans, important factors of the synovial fluid. In addition, variable leukocyte populations and blood vessels are present in the synovial tissue. In chronic inflammatory arthritis such as RA and PsA, the synovial tissue is the central site of inflammation, leading to synovial hyperplasia and local activation of synovial cells over time and, finally, resulting in irreversible cartilage and bone damage [2,3,4]. Synovial inflammation is characterized by increased permeability of the vessels, leading to leukocyte infiltration. Local proliferation and activation of RASFs is a typical and early feature of RA and, together with an increased influx of monocytes, differentiating into RA synovial macrophages, leads to the formation of the hyperplastic synovial lining layer and an increased cellular density within the sublining [2,3,4,13]. Due to the interaction of the different cell types and the high amounts of adipokines present within the synovial fluid and tissue, many efforts to evaluate the effects of adipokines on the different effector cells in the pathophysiology of RA have been made.

#### 3.1.1. Adiponectin in Rheumatoid Arthritis

Increased levels of adiponectin have been shown in the synovial fluid of the affected joints in RA [12,14,15]. Systemic adiponectin levels have been described to be increased in RA as well as associated with disease activity and disease progression [12,24,46]. However, not all studies have confirmed the correlation with disease activity [24,47]. This is most likely due to the different methods of normalization but also due to the heterogeneity of the disease. However, most studies have shown a correlation of adiponectin with inflammation markers such as C-reactive protein (CRP), including a recent study showing that total and HMW adiponectin are positively correlated with CRP levels [47]. A positive correlation between serum adiponectin and the disease activity score disease activity score (DAS)28 - ESR in RA patients was shown in another cross-sectional study [48]. Interestingly, subcutaneous adipose tissue from RA patients produced more adiponectin compared to OA patients and correlated with disease activity and disease duration in these patients [49]. Along this line, an increased risk for obese subjects with high serum adiponectin levels at baseline to develop RA, specifically with high adiponectin and CRP levels, was recently described in a study during a follow-up for up to 29 years [50]. The response to anti-TNF-alpha treatment seems to be associated with better improvement in patients with higher adiponectin levels at baseline [51]. Another study showed increased adiponectin and reduced chemerin levels in RA patients after anti-IL-6 treatment in both monotherapy or combined therapy with methotrexate independent of the treatment response [51]. Another recent study showed that tocilizumab treatment of RA patients with active RA despite previous conventional synthetic disease-modifying antirheumatoic drug (DMARD) and/or biological DMARD treatment was associated with an increase in total and HMW adiponectin, especially early after treatment onset, then declining until month 6 to 12 [52]. In this study, anti-IL-6 treatment also induced a gain in lean mass, while fat mass remained unchanged. Baricitinib, a JAK inhibitor blocking central inflammatory signaling pathways, was recently shown to decrease systemic inflammation biomarkers, such as IL-6, CRP, as well as adiponectin, in rheumatoid arthritis patients [53].

On the cellular level, numerous studies showed that cultured RASF respond to adiponectin by induction of pro-inflammatory factors, including prostaglandin E2, IL-6, IL-8, as well as matrix-metallo proteinases such as MMP-1 and MMP-13 [12,24]. Specifically, the HMW adiponectin isoforms seem to display stronger pro-inflammatory effects in RASF. Adiponectin stimulation of RASF indirectly affected T follicular helper cells, which was mainly mediated by the induction of IL-6 [54]. HMW adiponectin induced the production of IL-6 in unstimulated but not LPS-activated human monocytes, while LMW reduced IL-6 and increased IL-10 in LPS-activated monocytes [55]. Adiponectin seems to promote the differentiation of naïve T cells into Th17 cells, contributing to synovial inflammation and bone erosions, which may be mostly dependent on AdipoR1, demonstrated in conditional AdipoR1 knockout models [56]. Other cell types such as chondrocytes, endothelial cells, and lymphocytes also showed a pro-inflammatory response to adiponectin [12,24], suggesting a mainly pro-inflammatory effect of adiponectin locally within the affected joints (Figure 1). It was shown that adiponectin is able to increase the interaction of RASF with endothelial cells—for example, a cell-to-cell binding assay [57].

Fibroblasts are central cells in wound healing promoting angiogenesis, and increased angiogenesis is a prominent feature of RA. In addition, different bone cells also respond to adiponectin in RA, but this has also been shown for OA [15]. In osteoblasts, adiponectin induced pro-inflammatory cytokine production, which was reduced by targeting MMW/HMW adiponectin in osteoblasts but also ameliorated collagen-induced arthritis (CIA) in mice, showing a central contribution of those adiponectin isoforms to inflammatory processes in RA [54]. Therefore, different cell types respond to altered local adiponectin concentrations within the affected joints of RA patients. Besides central cells of cartilage and bone erosion, inflammatory cells are altered, showing different reactivity to the different adiponectin isoforms, representing an interesting approach to target adiponectin-mediated effects.

#### 3.1.2. Leptin in Rheumatoid Arthritis

Increased systemic leptin levels in RA have been discussed to be related with disease progression and disease activity [12,14,58,59]. Systemic leptin concentrations were shown to correlate with body fat percentage in RA patients [46,60], and an association between systemic leptin and IL-6 levels with cardiovascular risk of RA patients was reported [61]. In RASF and OASF, leptin induced the expression of IL-6 and IL-8 involving the JAK2/STAT3 and other signaling pathways. In leptin-deficient mice, the severity of arthritis was shown to be reduced together with IL-1beta and TNFalpha levels [14]. Interestingly, high levels of systemic leptin and vaspin were identified in early RA compared to healthy controls [62]. However, another study showed increased leptin levels in metabolic syndrome as well as patients with spondyloarthritis but not in patients with RA after one year of treatment with DMARD [63], and a decrease in disease activity correlated with decreased leptin levels could be observed in this study. In contrast, another study addressed RA patients with long disease duration (≥5 years), showing that pro-inflammatory markers, such as TNFalpha as well as resistin and leptin, were highest in long-duration RA, although they also increased in short-duration RA (<1 year) compared to healthy controls [64]. Therefore, the disease duration may have to be taken into account when evaluating leptin levels in RA.

Interestingly, an association with higher systemic leptin levels in the presence of periodontitis and specifically in individuals with increased markers of periodontitis was observed in patients with early RA [62]. Regarding treatment response, a short-term effect of tocilizumab was reported with respect to leptin serum levels in RA patients in a recent study showing a reduction of leptin following tocilizumab infusion [65]. Along this line, it was shown that tocilizumab but not methotrexate treatment increases body weight as well as the leptin–adiponectin ratio in RA patients in a retrospective cohort study [66]. Furthermore, a significant association of cardiovascular disease was reported in patients with RA, suggesting leptin to be a reliable prognostic factor and biomarker for predicting cardiovascular complications in RA patients [67]. A multi-biomarker disease activity score was used in another recent study to evaluate its ability to predict sustained remission in RA, showing that its biomarkers IL-6, serum amyloid A (SAA), CRP, as well as leptin can differentiate between small differences in disease activity and are also predictors of one-year remission [68]. Leptin represents a molecule with complex functions in metabolism. However, due to its association with comorbidities and treatment response, its potential as biomarker to monitor and compare treatment response in the context of inflammatory diseases is of high interest and is reflected in the high number of studies using leptin as an outcome parameter.

#### 3.1.3. Visfatin in Rheumatoid Arthritis

Pro-inflammatory responses in RASF, lymphocytes, monocytes, chondrocytes, and bone cells have been shown for visfatin in RA by different groups in the past years [12,15]. Visfatin levels are increased in RA compared to healthy controls and OA patients, and a correlation between systemic visfatin and inflammatory markers such as CRP and disease activity has been shown [12,24,60]. In contrast, visfatin expression seems to be associated with an adverse cardio-metabolic risk in RA, showing increased MMP-2 expression in relation to visfatin [69]. The potential of visfatin as a therapeutic target has been discussed for RA as small-molecule visfatin inhibitors are available and have already been attained in oncology [70,71].

Interestingly, collagen-induced arthritis (CIA) in visfatin-deficient mice showed reduced bone destruction, disease progression, and inflammatory activity in these animals [72]. In addition, this recent study showed that visfatin is required for osteoclastogenesis and that this requirement is most likely due to its Nampt activity. This is in line with another study using the APO866, an inhibitor blocking the enzymatic Nampt activity, leading to reduced CIA severity and production of inflammatory factors [29]. Regarding RASF, visfatin was recently shown to increase RASF adhesion to endothelial cells under static as well as flow conditions [57], potentially promoting angiogenesis and vessel guiding in RA synovial tissue. Visfatin is one of the adipokines with potential as a biomarker but also affecting different cell types in the affected joints in arthritis. Although the cell surface receptor for visfatin remains unclear, the enzymatic Nampt activity, which can be targeted using specific inhibitors, makes visfatin one of the most interesting adipokine targets in chronic inflammatory rheumatic diseases.

#### 3.1.4. Resistin in Rheumatoid Arthritis

Altered systemic resistin levels in the context of RA remain controversial [14]. However, a recent study showed highest levels of resistin in long-duration RA, but resistin levels were also elevated in short-duration RA compared to healthy controls [64]. Therefore, resistin levels may depend on RA disease duration and severity. Resistin is expressed in synovial tissue as well as in synovial fluid of RA patients compared to OA [14,73]. Increased synovial resistin seems to be correlated with disease activity and inflammatory parameters such as IL-6 levels and leukocyte count [14,32]. Several studies reported systemic resistin levels to be correlated with inflammatory biomarkers such as CRP, erythrocyte sedimentation rate (ESR), or TNFalpha [58,73,74]. However, other studies could not show a correlation of synovial fluid resistin with systemic CRP levels or systemic resistin levels with inflammation parameters [32,73]. On the other hand, intra-articular injection of resistin in a mouse model induced inflammation and hyperplasia in the synovial tissue, similar to arthritis [14,32]. Locally, stromal cells, such as RASF and osteoblasts, and immune cells, such as macrophages and B cells, express resistin in the affected RA joints [12,15,73]. Inflammatory cells such as human macrophages, synovial fluid leukocytes, or peripheral blood mononuclear cells (PBMC) respond in a pro-inflammatory manner to resistin [14,32] and so do RASF, specifically with the release of pro-inflammatory factors and chemokines [75]. Interestingly, anti-TNFalpha therapy rapidly reduced resistin serum levels in RA patients in close association with CRP [76]. Along this line, resistin expression was reduced in CD4 T cells and CD14 monocytes in RA patients responding to anti-TNFalpha therapy in contrast to patients who failed therapy response [77]. It was reported recently that after tocilizumab treatment for 24 weeks, resistin levels were significantly increased [66,78]. However, regarding anti-IL-6 treatment with tocilizumab of patients with active RA despite previous csDMARDS and/or bDMARDs, no significant changes in systemic resistin along with leptin and ghrelin during follow-up after 12 months were observed in contrast to adiponectin [52]. In addition, resistin was found not to be associated with the metabolic syndrome (by National Cholesterol Education Program’s ATP III and clinical parameters) at baseline and after one year of treatment of RA patients with DMRDs [63]. However, the decrease in resistin correlated with a decrease in disease activity in these patients [63]. It remains unclear whether the impact of disease activity may be more prominent compared to the metabolic syndrome in this treatment approach using DMARDs. In summary, the role of resistin as a biomarker and its pathophysiological role in the treatment response of RA patients are not fully understood and may depend on treatment approach, start of treatment within the course of the disease, as well as dosage.

#### 3.1.5. Chemerin, Vaspin, and Omentin in Rheumatoid Arthritis

Cleaved chemerin isoforms are increased in RA synovial fluid [34], and elevated chemerin plasma levels are correlated with BMI and disease activity in RA [79,80]. Recently, it has been postulated that visfatin and chemerin are increased in RA patients and chemerin can be used as an inflammation marker for RA patients [81]. Interestingly, total chemerin was found to be increased in RA serum, although it was recently revealed using a proteomics approach that mainly 155A, 156F, and 157S chemerin isoforms are present in RA serum, which is different from the distribution in polycystic ovary syndrome, also showing increased total chemerin levels [79]. A reduction in chemerin levels after treatment of RA patients with the anti-IL-6 inhibitor tocilizumab was observed [51], showing its potential as an inflammatory biomarker. Locally, chondrocytes and RASF express chemerin as well as its receptor [34,36,44]. Stimulation of RASF and OASF with chemerin induced IL-6, chemokines, and MMP-3 in these cells, and IL-1β in chondrocytes. In addition, chemerin induced RASF and leukocyte migration [44,45].

Increased systemic vaspin levels have been described to be associated with inflammation in RA [14,76]. In synovial fluid, vaspin levels were higher in RA compared to OA, but no correlation with inflammatory markers was observed [82]. Although the role of vaspin in the cells of RA pathophysiology is limited, there is some evidence that vaspin may affect different cells in RA development. However, it has been described that human osteoblasts are protected from apoptosis by vaspin, and vaspin was shown to inhibit osteoclastogenesis and osteoclast activity using a murine RAW264.7 macrophage cell line [14]. Increased vaspin levels have also been recently described for PsA patients compared to healthy controls [83]. It was shown in this study that the levels of vaspin and neutrophil gelatinase-associated lipocalin as well as the apolipoprotein B1/A1 ratios were significantly higher in PsA compared to controls, but none of the factors were correlated with disease activity [83]. In ankylosing spondylitis, low vaspin levels were found to be related to endothelial dysfunction [84], showing similar regulations and functions in inflammatory arthritis. However, studies evaluating the role of vaspin in the context of rheumatoid arthritis are limited.

Omentin was found to be lower in RA in synovial fluid compared to OA patients [82]. However, an association of systemic omentin with CRP has been described for RA patients [85]. In contrast, no elevated omentin concentrations were observed in RA tissues compared to OA [86]. This is in line with a study showing a weak response of both RASF and OASF to omentin stimulation, suggesting only weak effects of omentin in RA and on other effector cells in response to omentin [86]. However, serum omentin was also found to be increased in PsA compared to healthy controls [87].

Chemerin and vaspin showed a positive correlation with inflammation and, in part, disease progression. However, additional studies are required to fully assess the potential of these adipokines as biomarkers for disease progression and treatment response in RA. Furthermore, the effects of these adipokines on the cellular level require additional evaluation to fully estimate their role in chronic inflammation and contribution to cartilage and bone destruction in RA.

#### 3.1.6. Progranulin, Nesfatin, and Lipocalin-2 in Rheumatoid Arthritis

Progranulin (PGRN) levels were found to be increased in the synovial fluid and serum of RA patients compared to OA patients and healthy donors [14]. A correlation between systemic PGRN and disease activity and RA progression has been described [14,88], and thus PGRN may represent a promising marker for inflammatory responses in RA. Interestingly, the presence of anti-PGRN antibodies in RA patients was recently found to be associated with a higher disease activity compared to anti-PGRN-negative patients [89]. In addition, PGRN is an important mediator in the maintenance of cartilage integrity [14]. A protective effect on osteoblast differentiation under inflammatory conditions was also shown [90]. The observed increased PGRN concentrations in RA may, however, not suffice to inhibit the pro-inflammatory effects under chronic autoimmune conditions [40]. This is in line with findings in animal models of inflammatory arthritis, showing that reduced PGRN expression promotes more severe disease [41], and atsttrin, an engineered protein composed of three progranulin fragments, was shown to have similar effects to progranulin with beneficial effects in arthritis but apparently also anti-neuroinflammatory effects [91]. Interestingly, progranulin autoantibodies were detectable in approximately 25% of seropositive RA patients and in 21% rheumatoid factor and ACPA-negative RA patients [89]. The mean DAS28 values were significantly higher in progranulin autoantibody-positive RA patients in this study.

Nesfatin was found to be positively associated with rheumatoid factor in RA patients, correlated with MMP-2 concentrations and reduced atherosclerosis in these patients [69]. However, nesfatin-1 was not found to be associated with BMI or disease activity of RA patients in a recent study [92].

Lipocalin-2 is elevated in RA synovial fluid compared to OA patients [14,93,94] as well as in the serum of RA patients [95]. LCN2 is induced in joint tissues due to mechanical loading and inflammatory factors [45,93,94] but also by adipokines, such as leptin and adiponectin in chondrocytes [14,45]. TNFalpha and IL-17 are able to induce LCN2 in osteoblasts [14], and LCN2 induced in neutrophils by granulocyte macrophage colony-stimulating factor (GM-CSF) was associated with synovial cell proliferation and cell infiltration into RA synovial tissue [94]. In ankylosing spondylitis, elevated lipocalin-2 values were found, and in the ank/ank mutant mice, a mouse model for ankylosing spondylitis, increased lipocalin-2 levels were associated with the coexistence of ankylosis and gut inflammation observed in these animals [96].

Additional studies evaluating adipokines such as lipocalin-2 and nesfatin are required to elucidate the potential of these factors as biomarkers and immunomodulatory proteins. In contrast, several studies addressing the potential of progranulin as an inflammatory marker in the context of RA showed a clear correlation with disease activity in humans as well as in animal models of inflammatory arthritis. Therefore, PGRN represents an interesting adipokine with potential as a biomarker, as well as an immunomodulatory protein affecting different cell types.

#### 3.1.7. Adipokine Network in Rheumatoid Arthritis

Many studies evaluate several adipokines, contributing to the knowledge of cross-talking adipokine networks and relationships between adipokines as molecular mediators in RA. Systemic chemerin and visfatin were both increased in RA compared to healthy controls with chemerin, showing the highest specificity for RA [81]. On the cellular level, both visfatin and resistin alter the expression of miRNA in chondrocytes, leading to both reduced viability and increased apoptosis [30], thus showing the role of these adipokines in cartilage erosion. Regarding the relationship of adiponectin and leptin, serum leptin and adiponectin normalized to body fat mass were both significantly higher compared to healthy controls, and adiponectin concentrations correlated positively with the degree of bone destruction and serum MMP-3 in contrast to leptin [48], suggesting a relationship of adiponectin with bone erosion and both adipokines with inflammation. However, another study found both systemic leptin and resistin levels to be correlated with inflammatory markers and RA activity independent of the subjects’ BMI in contrast to adiponectin when comparing patients with low, moderate, and high disease activity according to DAS28 [74]. In RA patients with active disease, treatment with tocilizumab significantly reduced chemerin, while increasing adiponectin, and leptin and resistin were not altered in this study [51]. However, another study addressing tocilizumab treatment of RA patients found significantly higher adiponectin levels in RA at baseline, while after tocilizumab treatment, resistin levels and the leptin–adiponectin ratio were increased; in contrast, BMI-adjusted adiponectin levels were decreased and leptin levels remained unchanged [78]. These studies show that adipokine adjustments have to be taken into account besides disease activity to clearly elucidate the role of adipokines and adipokine relationships systemically. However, these and many previous studies clearly show that adipokines are altered in the context of chronic inflammatory diseases and represent biomarkers of inflammation as well as disease activity, joint erosion, and treatment response.

#### 3.1.8. Summary and Conclusions

Chronic inflammation leads to tissue damage, which, in case of chronic inflammatory joint diseases, results in irreversible damage of cartilage and bone. Long-term pain and loss of locomotor function have a negative impact on the health care and social system. In the past years, it was shown that metabolic factors, specifically factors produced in high amounts by adipose tissue and adipocytes, contribute to inflammatory processes not only in RA but also in many other autoimmune diseases. Many autoimmune diseases are very heterogeneous, leading to the need for different treatment options. Finding common markers to allow a better and early identification as novel biomarkers and diagnostic tools, including evaluation of treatment efficiency, would be of high value. In addition, it is well established that not only protective effects are mediated by adipokines but that they also contribute to inflammatory responses on immune cells. Besides, connective tissue cells and in RA bone cells are altered by adipokines such as adiponectin and visfatin. Adipokines are able to interact and in part induce each other, forming a complex adipokine network. Due to partially anti- as well as pro-inflammatory functions of some adipokines, a specific role is difficult to assess in a complex and multifactorial disease such as RA. The high interest in evaluating the role of adipokines in synovial inflammation as well as cartilage and bone erosion, leading to progressive destruction of affected joints, is visible in the number of studies performed in the past years, which contributes to our knowledge regarding the role of adipokines in autoimmunity as well as specifically in the pathophysiology of chronic inflammatory arthritis.

## Figures and Tables

**Figure 1 cells-10-00216-f001:**
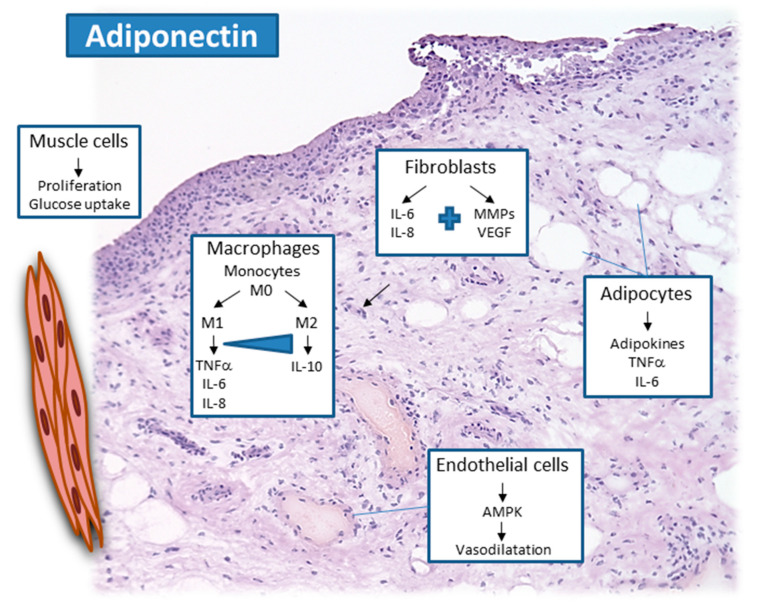
Adiponectin activates local synovial cells, such as synovial macrophages and synovial fibroblasts in the lining layer and sublining, as well as endothelial cells and adipose tissue and muscle in the deeper and adjacent areas of the joint.

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
