# Peer review of "Adipokines and Autoimmunity in Inflammatory Arthritis"

_cells, 2021, doi:10.3390/cells10020216_

Round 1

Reviewer 1 Report

The manuscript entitled "Adipokines and Autoimmunity in Arthritis" reviews the recent literature focused on the role played by different of the most relevant adipokines in rheumatoid arthritis. The review is updated and shows recently identified effects of "classic adipokines" as well as different functions exerted by newly identified adipokines.

I only have a minor suggestion:

Line 159: I believe that the authors meant resistin instead of leptin.

Author Response

Dear Reviewer,

Thank you for your time and suggestions. Below please find a point-by-point answer to the raised points.

  1. Line 159: I believe that the authors meant resistin instead of leptin.

Answer: The mistake was corrected and leptin replaced with leptin.

Reviewer 2 Report

The authors aim to address the most recent findings on the role of adipokines on RA. They place RA in the context of joint disorders and autoimmune diseases and discuss several adipokines separately. Unfortunately, the aim of describing the role of adipokines as a network of interactions in the pathophysiology of inflammatory arthritis is not met. No interactions between the different adipokines are discussed and no larger viewpoint is presented. Also, the manuscript could be edited so that it is more clear and gives a more comprehensive overview. I have made a number of suggestions below on how that could be achieved.

The manuscript would benefit from language editing. Especially the use of commas and conjunctions is causing a lack of clarity. For instance, in line 41-42 and line 59-60.

Furthermore, the individual sections of the manuscript lack focus and the abstract does not give a comprehensive overview of the manuscript (like the conclusion does). The manuscript would also benefit from more figures and a summarizing table.

The separate introductions on each adipokine make the manuscript very difficult to read. I would suggest shortening the introduction to only place RA in context of arthritic diseases and autoimmune diseases and then introduce why adipokines are of interest (I liked the beginning of the conclusion section ). The main body should start with the details on RA and then each adipokine should be introduced and in the same section, its role in RA should be discussed and a summarizing conclusion for each adipokine should be provided (similar as the one provided for resistin). It is unclear to me why the references are included in the current order and which part is introductory (using reviews to make a general point on the role of an adipokine in RA) and which are recent original research studies. It would help if a little more background for each referenced article would be included(such as type of study or study population, see also below). Now, it is necessary to look up all the references and read them to understand what the relevance and context is and it is difficult to keep the overview. It would be nice if this review does this job for the reader. Possibly, providing separate paragraphs on the role as biomarker, responses to treatment and inflammatory effects (with clear distinction between in vitro, animal models and patient findings) would help with this as well.

At the end, the network interactions between the different adipokines should be discussed.

Finally, the conclusion section reads more like the abstract, than a conclusion. It starts from the beginning and the statements are somewhat contradictory to those in the introduction where it is stated that inflammation is a result of cartilage damage, not the cause. Here, for the first time it is clearly stated that the adipokines can be used as biomarkers, which would have placed many of the reviewed findings in the necessary context.

Some minor comments:

Line 90-96 not clear why this is important.

Line 101-103 please be more specific and give details.

Line 218. Do you mean “limited availability”?

Line 300 and 308 the same article is referenced but the relationship is not clear. Also, any explanation or details that may indicate why the difference between refs 63, 64 and 65 occurs would be helpful (age of patients, location, aim of study etc). t

Line 320-321. Both articles you refer to are review papers and one is from 2011. Please include original references (“from the past years”). The same for lines 342-344, and lines 403-405.

Line 357-358. Please make more clear that this is the unexpected finding and include some details from the study that indicate its importance. As most other reported findings indicate an association with disease severity, this seems to be the only controversial finding that is included in this section. Now, that detail is easily missed and impossible to put in context without looking up the article.

Line 352-353: “as well as” should be replaced by “and so do”. Otherwise you are saying that immune cells respond to RASF.

Line 361-362. It would be interesting to know what criteria were used to define metabolic syndrome in these patients (ref 64). Could it be that RA disease activity has a larger effect on resistin levels than metabolic syndrome and that is why no association is found? It is unclear how this reference is related to the response to treatment that is mentioned in the previous and next sentences.

Reviewer 3 Report

Dear authors,

This manuscript ‘Adipokines and Autoimmunity in Arthritis’ is a very informative, insightful paper that extensively reviews (chiefly the pro-inflammatory aspect of) adipokines in rheumatoid arthritis. I have the following comments:

Line 209. Adipokines in autoimmune rheumatoid arthritis; removing ‘autoimmune’  seems more appropriate.

Page 8 Figure 1. The background of showing synovial tissue does not perfectly fit with the multiple facets depicted in the figure. I understand it would be difficult to incorporate all tissues in one figure. Illustrations of each cell or tissue, respectively, under each group, may a better option.

(Optional) A table summarizing the role of respective adipokines in rheumatoid arthritis (under 3. Adipokines in autoimmune rheumatoid arthritis) could facilitate a better understanding of your message. 

Title The title is straightforward and concise, though the concept of ‘Arthritis’ is fairly broad. Moreover, the differential of inflammatory arthritis is wide, and this paper does not dive into each entity. Therefore, the title ‘Adipokines and Autoimmunity in Rheumatoid arthritis’, or ‘Adipokines and Autoimmunity in Inflammatory arthritis’ seems more feasible. 

Round 2

Reviewer 2 Report

The manuscript has improved significantly and is suitable for publication.